# 1-Tetradecanol, Diethyl Phthalate and Tween 80 Assist in the Formation of Thermo-Responsive Azoxystrobin Nanoparticles

**DOI:** 10.3390/molecules27227959

**Published:** 2022-11-17

**Authors:** Guan Lin, Juntao Gao, Shenghua Shang, Huanbo Zhang, Qiangke Luo, Yutong Wu, Yong Liu, Xingjiang Chen, Yan Sun

**Affiliations:** 1School of Biological & Chemical Engineering, Zhejiang University of Science & Technology, Hangzhou 310023, China; 2Guizhou Academy of Tobacco Science, Guiyang 550001, China; 3Qianjiang College, Hangzhou Normal University, Hangzhou 310018, China; 4Zhejiang Provincial Key Laboratory for Chemical and Biological Processing Technology of Farm Product, Hangzhou 310023, China

**Keywords:** nanoparticle, 1-tetradecanol, thermo-responsive, azoxystrobin, bioavailability, efficacy

## Abstract

The occurrence of crop fungal diseases is closely related to warm environmental conditions. In order to control the release of fungicides in response to warm conditions, and enhance the efficacy, a series of thermo-responsive fungicide-loaded nanoparticles were developed. The fungicide azoxystrobin, solvent DEP, emulsifier Tween 80 and thermo-responsive component TDA were combined to create thermal-response oil phases, conditions for emulsification were then optimized. LDLS, zeta potential, FTIR, DSC, TGA, XRD, SEM and antifungal efficacy assays were carried out to investigate the characteristics and forming mechanism. The results indicated that the formula with 5 g azoxystrobin, 10 mL DEP, 6 mL Tween 80 and 2.5 g TDA constructed the proposed oil phase with the ability to transform from solid at 20 °C to softerned at 31.5 °C. Both DEP and TDA played key roles in interfering with the crystallization of azoxystrobin. The optimal T3t-c12 nanoparticles had a mean particle size of 162.1 nm, thermo-responsive morphological transformation between 20 °C and 30 °C, AZO crystal reforming after drying, the ability to attach to fungal spores and satisfied antifungal efficacy against *P. nicotiana* PNgz07 and *A. niger* A1513 at 30 °C. This report provides referable technical support for the construction of smart-release nanoparticles of other agrochemicals.

## 1. Introduction

According to the United Nations (UN) Food and Agriculture Organization and the Organization for Economic Cooperation and Development, the agricultural sector is fast growing due to high requirements for the raw crop materials necessary for the development of new applications, including biofuels [1,2,3]. However, fungal infections in plants have caused serious threats to food supply and security over the past decade. Approximately 20% of global crop yield is lost to phytopathogenic fungi annually [4]. For this reason, agricultural fungicides have been used for centuries to protect plants from disease. In addition, many conventional fungicide formulations have been observed to have several drawbacks, including low bioavailability, low water solubility, high volatility and high susceptibility to leaching [5]. The active ingredients in fungicides have a difficult time reaching their targets; hence, they are lost to the environment, resulting in leaching and environmental pollution [6,7]. Research has therefore sought to develop formulations that can circumvent these challenges by advancing a wide range of technologies, one of which is to minimize the size of particles or droplets to the nanoscale order. Nanoparticles (NPs) and nanoemulsions (NEs) have uniquely small dimension effects and have received immense attention in several fields, including the biomedical, food, cosmetic and pharmaceutical industries [8]. The nanometerization of fungicides results in a high possibility of increases in wettability, spreadability and mechanical stability in comparison to common suspensions because of the smaller droplet sizes. These characteristics may aid in reducing volatilization and the hydrolysis of the active ingredient, thus augmenting bioavailability [7,9,10]. Nanoformulations of azoxystrobin (NPs and NEs), pyraclostrobin (NPs and NEs), hexaconazole (NPs), propiconazole (NPs), prothioconazole (NPs), mancozeb (NEs) [11], fenoxanil (NPs) [12], fosetyl-Al (NPs) [13], nano-AgCu alloy (NPs) [14] and ZnO (NPs) [15] et al. have been reported. Regardless of the kinds of coating materials and loading strategies devised and actualized for fungicide delivery, all of these positive results point in the same direction: using nanotechnology to create novel nanoformulations shows promise in improving the efficacy and safety and reducing the frequency of use of agrochemicals, including fungicides [16,17,18].

Azoxystrobin (AZO) is a systemic methacrylate fungicide that is derived from the strobilurin family. As the first globally available germicide in the past decade, AZO attracted well-deserved attention on the development of its novel nanoformulations. Minimizing the size of AZO crystals to nanoscale is a popular research objective. Yao, J., et al. prepared a nanosuspension of AZO crystals using a wet media milling method [19]. Camiletti et al. employed wet bead milling followed by a spray-drying procedure to generate self-dispersible AZO nanocrystals [20]. Both AZO NPs prepared by Yao and Boris presented particle sizes below 300 nm and remarkably improved the performance of the fungicide. Currently the smallest AZO nanosuspension, with a mean particle size of 84 nm, is that created through a flash precipitation method by Zhu, Z., et al. With this nanosuspension, higher biological efficacy against *A. niger* was observed compared with those made via a traditional precipitation method and the Amistar nanosuspension from Syngenta [21]. In addition to the preparation of nanoscale AZO crystals, a series of AZO-loaded composited NPs also were reported. Among these, encapsulation of AZO in porous hollow silica NPs is the widely accepted and successful method. Xu, C., et al. prepared an AZO-loaded nanoemulsion with a combination of inorganic mesoporous silica and organic carboxymethyl chitosan. The AZO-loaded NPs exhibited 477 nm and 222 nm hydrated and dried particles, respectively. At the same time, the organic–inorganic composite AZO-loaded NPs also exhibited a pH-responsive release profile and better fungicidal activity against tomato late blight *P. infestans* than AZO alone under the same doses of active ingredient applied [22]. An AZO-loaded pH-response releasing nanoparticles (AZOX@MSNs-PDA-Cu) was also created by Xu, C., et al. on the base of mesoporous silica, dopamine chemistry and copper ions chelate complex [23]. The AZO-encapsulated porous hollow silica nanoparticles made by Bueno et al. showed high loading efficiency for azoxystrobin, controlled release, better plant growth than nonencapsulated AZO and few to no changes to the soil microbial community [24,25]. Besides silica base carriers, AZO has also successfully been embedded in rationally designed amphiphilic plant oil-based bottlebrush copolymer [26], tannic acid-based nanopesticides [27], chemically cross-linked lignin nanocarriers [28] and layered double hydroxide nanosheets [29].

Based on the analysis of the molecular structure, it can be found that AZO is highly susceptible to crystallization. Every AZO molecules contain 2 strong non-polar phenyl groups and 1 strong polar nitrile group, which makes it easy to form an ordered molecular alignment in mixture of common organic solvent and water. So, how to suppress the continuous growth of crystals proves to be a hard question after we have recognized the strong crystallization power of AZO. We believe that the optimal combination of solvent and emulsifier and the construction of space obstruction to inhibit crystal growth is a promising idea for the preparation of AZO NPs. In our previous work, the combination of Diethyl phthalate (DEP) and Tween 80 (T80) was found to be suitable for forming a stable AZO nanoemulsion. After the further combination of a thermal responsive component, for example, 1-cetanol (CTA), 1-tetradecanol (TDA) or 1-dodecanol (DDA), AZO-loaded thermo-responsive NPs might be obtained. This paper introduces the main research procedure and results surrounding the development of novel thermo-responsive AZO-loaded NPs. 

## 2. Results

### 2.1. Combinations of TDA, T80 and DEP Affect the Softening Points of Oil-Phase Mixtures

The softening points of CTA1, TDA1 and DDA1 were determined as 49.4 °C, 37.5 °C and 22.7 °C, respectively. Thus, TDA was considered the most beneficial thermal responsive component for AZO-loaded NPs. The softening points and endothermic peaks in the DSC of samples with different TDA and DEP additive amounts are shown in Table 1. The thermal behaviors of the oil-phase samples during DSC and TGA analysis are shown in Figure 1. The results in Table 1 show that the softening points descended step by step as the addition of TDA was gradually decreased from T1t to T4T. On the other hand, a decrease in DEP additive amounts (comparison between T1t, D1t and D2t shown in Figure 1) and a decrease in both the DEP and TDA amounts (comparison between T1t and T5t shown in Figure 1) caused an increase in softening point. The thermal behaviors of the oil-phase samples (Figure 1) showed the separation of a TDA melting peak at 37.8 °C and AZO melting peaks at 98.6 °C and 110.2 °C for CK2n, a DEP and Tween 80 dual absent sample, indicating the incompatibility of AZO and TDA; a comparison between CK2n and CK2t indicated that the addition of T80 weakened the AZO melting peak at around 111.6 °C and decreased the melting temperature of TDA from 37.8 °C to 34.7 °C. The addition of DEP increased the melting peak of TDA to 34.7 °C in CK2t, which was split into an accompanying peak at 31.1 °C and recovered at 37.8 °C in T1t. The doping of the unsaturated T80 alkane chain and DEP with TDA saturated alkane chain should interfere with the crystallization behavior of TDA, and finally form a partially crystalline and partially amorphous solid with a low softening point. The continuous weakening and movement toward the low-temperature direction of the endothermic peak at 37.8 °C in T1t, 34.3 °C in T2t, 33.7 °C in T3t and 31.1 °C in T4t showed a strong positive correlation with decreasing TDA amounts in the oil-phase mixtures. As a solid fatty alcohol and phase change material (PCM), TDA has two phases: the metastable hexagonal orthorhombic solid phase (S_HEX_) and the orthorhombic solid phase (S_ORT_) [30,31]. The AZO-constructed porous network and shock-cooling treatment (direct transfer from 55 °C to 0 °C ice/water bath) should hinder the phase transformation of TDA from S_HEX_ to S_ORT_, generating the defective crystals with the low melting point [32]. Therefore, The continuous decreases of the softening points should contribute to the decreasing crystallinity of the oil-phase mixtures due to decreasing TDA amounts. The disappearance of the AZO melting peak in T2t and T3t implied a well-reconstructed dissolving form of AZO, and the reappearance of the AZO glass transition temperature at 105.3 °C in T4t was attributed to the loss of the blocking effect against AZO crystallization caused by TDA insufficiency. The TGA results (Figure 1) indicate that components AZO, TDA and optimal oil-phase mixture T3t are thermally stable in the temperature range of DSC results. Thus, T3t should be an applicable oil-phase for the preparation of thermo-responsive AZO NPs.

### 2.2. TDA and DEP Play the Critical Role in AZO Crystallization in Oil-Phase Mixtures

The XRD patterns of the oil-phase samples (Figure 2) indicated that the typical diffraction peak at 10.8° in AZO was not observed in the oil-phase samples, except T4t. In addition, T1n, T1t, T2n, T2t, T3t and T4t exhibited highly similar XRD patterns to those of the AZO-free samples CK0n and CK0t. The addition of T80 did not generate any visible effect on AZO crystallization. The 105 °C glass transition temperature of T4t in DSC was consistent with the characteristic diffraction peak of AZO observed in XRD at 10.8°, which indicated that in the T4t oil-phase sample, AZO retained its original crystal structure to some extent. The FTIR results (Figure 3) also showed partial changes in the vibration of some unique groups. The stretching vibration of the phenyl C=C bond from DEP at 1580.9 cm^−1^ moved downward to 1561.4 cm^−1^, representing the C=C bond vibration from AZO, due to the AZO-induced co-frequency phenyl vibration of DEP (Figure 3A). According to decreasing DEP amounts (from T2t to D2t to D1t in Figure 3A), four variations in vibration peaks were also found: (1) The bending vibration of the phenyl group at 1606.1 cm^−1^ regained step by step, while it was submerged into the peak slopes in the DEP-rich samples T3n, T3t, T2n and T2t. (2) The stretching vibration of -CH2- groups at 1457.2 cm^−1^ became sharper and stronger. (3) The sharp vibration peak of -OH from the imidazole group on AZO at 997.7 cm^−1^ reformed stepwise in D2t, contrary to the wide blunt peaks in T3n, T3t, T2n and T2t. (4) The scissoring vibration of -CH_3_ at 1381.9 cm^−1^ moved to a higher wavenumber because of increasing spatial block, while it moved back on D2t. These variations indicate that DEP might work as a compatibilizer between AZO and TDA in the mixtures. The effect of TDA amounts on FTIR results is shown in Figure 3B. The stretching vibration of carbonyl at 1730.6–1708.9 cm^−1^ was overlapped, the stretching vibration of the -C=C- of AZO at 1626.7 cm^−1^ moved toward higher wavenumbers because of spatial block, and the peak shape became wider with a decrease in TDA amounts from T1t to T4t. The peak group around 1446.1 cm^−1^ presented a gradual switch from DEP type to AZO type, in keeping with the reduction in TDA amounts from T1t to T4t. The vibration of -C-O-C- in the ester linkage at 1258.1 cm^−1^ became separated stepwise from T1t to T4t. The bending vibration at 1155.9 cm^−1^ belonging to the diphenyl ether bond of AZO disappeared in all the AZO-DEP-TDA mixtures, regardless of whether T80 or low levels of DEP were added; thus, the doping of TDA mostly contributed to the fixation of AZO molecules.

### 2.3. Combination of Suitable Co-Efficients for Viscosity, Stirring Speed and Ionic Surfactant for the Creation of Stable AZO-Loaded NP Suspension

The mean particle sizes by intensity of the nanoparticle suspension prepared with T3t oil-phase mixture are shown in Figure 4 (an expanded figure of AZO-loaded particle sizes including T2t and T4t oil-phase mixtures is shown in Appendix A). The mean particle sizes of T3t-03 and T3t-13 were recorded as 191.2 nm and 193.7 nm, respectively. Thus, 0.1 ‰ (wt/v) xanthan gum was considered a preferred option because it helped to keep the suspension stable by inhibiting particle precipitation in addition to having an acceptable emulsification effect. The stirring speed for subsequent experiments was set at 10,000 rpm. The control treatments (CK0 and CK1) were prepared under the optimal 0.1 ‰ (wt/v) xanthan gum concentration and stirring at 10,000 rpm; mean particle sizes of 322.9 nm and 214.8 nm, respectively, were obtained. The particle sizes and zeta potentials of AZO-loaded NPs prepared with additional ionic emulsifiers, SDS and CTAB, are shown in Figure 5. The results indicated that SDS caused an increase in particle size, while CTAB acted to minimize the particle size of AZO-loaded NPs. T3t-c13 had a mean particle size of 109.2 nm at a 15 % AZO concentration (wt/v); however, it took on a non-fluid gel form after standing at room temperature overnight. The particle size, zeta potential and AZO concentration of T3t-c12 reached 162.1 nm, 5.1 mV and 10 % (wt/v), respectively. The fluidity of T3t-c12 was acceptable after standing at room temperature overnight. Thus, T3t-c12 was applied for further characterization.

### 2.4. Thermo-Responsive AZO-Loaded NPs Underwent Morphological Transformations at 30 °C and Original AZO Crystals Were Recovered after Drying

The thermo-responsive behaviors and crystal characteristics of T3T-13 and T3T-c12 under SEM observation are shown in Figure 6 and Figure 7, respectively. When there was no TDA in the nanosystems, the particle morphology stayed the same whether they were desiccated at 20 °C or 30 °C (Figure 6: CK0-20, CK0-30). For the TDA-containing samples, an obvious transformation in NPs appearance from regular to irregular spheres was observed when the desiccating temperature increased from 20 °C to 30 °C. The coverage area of the T3t-c12-30 NPs increased when compared with that of the T3t-c12-20 NPs; the positively charged surface may have driven their dispersal on the negatively charged silicon surface. Though T3T-13 and T3T-c12 showed different thermo-responsive behaviors, the reappearance of the typical AZO diffraction angle around 11° on dry T3T-13 and T3T-c12 NPs (Figure 7: T3T-13-NPs and T3T-c12-NPs) indicated that both recovered their original AZO crystal structures after the NPs were desiccated; this should benefit the original systemic activity of AZO.

### 2.5. Thermo-Responsive AZO-Loaded NPs Attached to A1513 Spores Resulted in Morphological Changes

The exterior morphology of A1513 spores with or without AZO-loaded NPs treatment is shown in Figure 8. Sharp and high thorn-like structures could be seen on the surface of pure A1513 spores without AZO-loaded NPs treatment (Figure 8: CK-20 and CK-30). After co-incubation with AZO-loaded NPs, the exterior appearance of the A1513 spores changed to short and blunt nodes on the surface instead of the original sharp and high thorns. The attachment of NPs to the A1513 spores likely contributed to this change in appearance. T3t-c12-20 had a lower superficial roughness than T3t-13-20, and T3t-c12-30 showed the lowest superficial roughness out of all six examined samples. The electrostatic attraction between the negatively charged A1513 spores and positively charged T3t-c12 NPs may cause more NPs to adsorb on the spore surface and eventually cover most of the thorns and lacunes. The thermo-responsive softening of T3t-c12 NPs at 30 °C made the spore surface look smoother.

### 2.6. T3t-13 and T3t-c12 Showed Satisfactory Antifungal Efficacy against A. niger A1513 Spores

The correlation curve of AZO concentration and suppression rate to A1513 spore germination (Rsa) is shown in Figure 9. T3t-c12 had a higher Rsa than that of T3t-13. In addition, the LC_50_ values for T3t-c12 and T3t-13 NPs were calculated according to the formulae in Figure 9. The LC_50_ of T3t-13 and T3t-c12 against *A. niger* A1513 spores was determined as 52.5 μg/plate and 40.8 μg/plate, respectively. The LC_50_ of T3t-c12 was 20.6% lower than that of T3t-13, which indicated that higher antifungal efficacy was obtained with the T3t-c12 NPs. 

### 2.7. T3t-c12 Showed Satisfactory Antifungal Efficacy against the Mycelium of P. nicotiana PNgz07

An evaluation of the efficacy and accumulative release rate of AZO in culture medium is shown in Figure 10. The mycelium of *P. nicotiana* PNgz07 grew normally on Tab-5 agar when no AZO was added to the medium (Figure 10: PN-20). The growth suppression rates (Rgs, %) of CK-0-20, T3t-c12-20 and T3t-c12-30 against *P. nicotiana* PNgz07 at an initial AZO concentration of 100 μg/mL were determined as 88.6 ± 4.7%, 71.5 ± 2.4 % and 72.6 ± 0.7%, respectively, while the Rgs of T3t-c12-30 was slightly higher than that of T3t-c12-20. However, the density of the aerial mycelium on the T3t-c12-20 plates was obviously greater than that on the T3t-c12-30 plates. The continuous decline in the accumulative rate of AZO in CK0-20 may be related to the dissipation of AZO, caused by multiple factors. AZO is susceptible to degradation by photolysis [33,34,35]. The sorption effect [36], leaching effect [37], and biodegradation [38,39,40] also contribute to a decrease in AZO under various circumstances. The coating of AZO with TDA should slow down the release of AZO and protect AZO from degradation before release; this was evidenced by the increase in the accumulative release rate of AZO in T3t-c12-20 and T3t-c12-30 medium during the first 4 d, followed by a decline in the following 6 d. The antifungal activity on T3t-c12-30 was achieved with the support of a relatively high free AZO level in the medium. The AZO standard curve used for determination under HPLC conditions is shown in Appendix A. 

## 3. Discussion

### 3.1. DEP Mediated the Miscibility between AZO and TDA

DEP is widely used as a plasticizer; it has a strongly hydrophobic phenylic group, while AZO has two phenylic groups. According to the mechanisms of action of plasticizers [41], both the “attractive forces to function” from lubricity theory and the “solvation-desolvation/aggregation-disaggregation” model from gel theory match the function of DEP in this AZO-loaded thermo-responsive NPs system. The movement of the stretching vibration of the phenyl C=C bond in DEP at 1580.9 cm^−1^ downward to 1561.4 cm^−1^ near the C=C bond vibration of AZO (Figure 3) supports the action of attractive forces or molecular inducement; in addition, the disappearance and reappearance of AZO melting endothermic peaks and the movement of TDA melting endothermic peaks in the AZO-DEP-TDA-T80 mixtures indicated the existence of AZO solvation and the partial aggregation of TDA molecules. Thus, DEP played an indispensable role as a compatibilizer in the AZO-loaded thermo-responsive NPs system.

### 3.2. Quick Solidification of TDA Suppressed the Growth of AZO Crystals

TDA is an eco-friendly natural product with a melting point at 37.7 °C [42] and as low as around 0.2 mg/L solubility in water [43] at 25 °C and 760 mm Hg atmospheric pressure. The melting point of DEP falls at −40.5 °C [44], while AZO possesses a softening point around 116 °C [45]. Under the guidance of “Gel Theory and Free Volume Theory of Plasticizer”, DEP mediates the interaction between AZO and TDA molecules, forming free volume among the TDA molecules [46]. When melting TDA transforms into its solid form in an ice/water bath, the TDA scaffold will immediately causes steric hindrance [47], and the growth of AZO crystals is interdicted. The following results from this study support this explanation: (1) the AZO melting endothermic peaks disappeared when TDA was added at higher levels and reappeared when the TDA amount was downregulated to lower levels (Figure 1); (2) the typical X-ray diffraction peak of AZO around 10.8° also disappeared and reappeared in the same way (Figure 2 and Figure 7); (3) TDA-associated melting appearances were observed in the AZO-loaded NPs (Figure 6 and Figure 8). Thus, the key role of TDA during the formation of stable thermo-responsive AZO-loaded NPs was shown.

### 3.3. Thermo-Responsive Property of Agrichemical NPs in Warm Temperature Regions Can Match the Requirements for Agricultural Application

The acceptable soil temperature and air temperature for planting and growing most vegetables are around 4–30 °C and 10–40 °C, respectively; however, in extreme climate conditions, the leaf temperature of maize, sugar beet and red clover can reach over 40 °C [48]. The occurrence of common fungal, bacterial and viral diseases in crops is highly correlated with environmental temperature and moisture conditions [49,50]. Thus, the responsive release of agrichemicals in a region with warm temperatures around 25 °C should be in line with these expectations. Unlike cold response releasing mold [51], agricultural applications require responsive release at warm temperatures and the holding of agrichemicals at cold temperatures. This thermo-responsive release property in warm temperature regions can be considered as the unique “Warm-Response Releasing (WRR)”.

## 4. Materials and Methods

### 4.1. Materials and Instruments

Diethyl phthalate (DEP), Tween 80 (T80), 1-cetanol (CTA), 1-tetradecanol (TDA), 1-dodecanol (DDA), sodium dodecyl sulfate (SDS), cetyltrimethylammonium bromide (CTAB) and xanthan gum were AR-grade products obtained from Aladdin Reagent Co. (Shanghai, China). Azoxystrobin (AZO, 96%) was obtained from Shanghai Macklin Biochemical Technology Co., Ltd. (Shanghai, China). Methanol, acetonitrile and acetic acid (HPLC grade) were acquired from Tedia Company, Inc. (Fairfield, CA, USA). K_2_HPO_4_, vitamin B1(VB1), glucose, MgSO_4_ and agar powder were obtained from Tianjin Zhiyuan Chemical Reagent Co., Ltd. (Tianjin, China). Yeast extract and tryptone were acquired from Oxiod Ltd. (Lancaster, UK). PDA medium and calf extract were procured from Hangzhou Microbial Reagent Co., Ltd. (Hangzhou, China). Potato powder was provided by Gansu Zhengyang Agricultural Science and Technology Co., Ltd. (Qingyang, China). Nutrient soil was provided by Chengdu Chunnian Horticultural Company (Chengdu, China). Both the Fourier Transform Infrared Reflection (FTIR) spectrometer Paragon 1000 and Differential Scanning Calorimetry (DSC) calorimeter Pyris 1 were from Perkin Elmer (Waltham, MA, USA). The Thermal Analysis System TGA 2 was from Mettler Toledo (Columbus, OH, USA). The X-ray Powder diffractometer (XRD) DX-2700 was from Haoyuan Instrument Co., Ltd. (Dandong, China). The scanning electron microscope (SEM) SU1510 used for analysis was from Hitachi (Ibaraki, Japan). The Particle Zetasizer 3000HS was from Malvern Instruments (Malvern, UK). The high performance liquid chromatography (HPLC) equipment (Waters e2695 with Waters 2489 UV detector and HPLC column: Sunfire^TM^ C18 (250 × 4.6 mm, 5 μm)) were from Waters Corporation (Milford, MA, USA). The asphalt softening point tester used (CHR-2806E) was from Hebei Shengruida Technology Co., Ltd. (Shijiazhuang, China). The autoclave was from Xiamen Zhiwei Company (Xiamen, China). The SYC-1025D water bath and OES blender were from Shanghai Qiuzuo Scientific Instrument Co., Ltd. (Shanghai, China). The plant pathogen strain *Phytophthora nicotiana* PNgz07 was isolated from diseased tobacco plants in Guizhou Province, China, while the *Aspergillus niger* A1513 strain was a model fungal strain stored in our lab.

### 4.2. Preparation of Thermo-Responsive AZO-Loaded NPs

#### 4.2.1. Screening for an AZO Oil-Phase Formula with a Softening Point Close to 30 °C

Five grams of AZO was dissolved in 10 mL of DEP at 75 °C under magnetic stirring, and 6 mL of T80 was added into the AZO-DEP mixture. Then, 10 g of CTA, TDA or DDA was added into the AZO-DEP-T80 solution to form clear oil-phase solutions. The oil-phase solutions were named CTA1, TDA1 and DDA1 because they contained CTA, TDA and DDA, respectively. The softening points of the three oil-phase formulas were tested on an asphalt softening point tester(CHR-2806E). After TDA was recruited as a thermal responsive component, samples with different amounts of TDA and DEP added were prepared as shown in Table 2. Then, the AZO powder and samples from Table 2 were characterized using FTIR, DSC and XRD to investigate the existing form of AZO in the oil phase.

#### 4.2.2. Optimization of Emulsification Conditions to Prepare Thermo-Responsive AZO-Loaded NPs

Hypothetical thermo-responsive AZO NPs were prepared by dispersing oil-phase T2t, T3t and T4t in xanthan gum water solutions under stirring. The xanthan gum concentrations and stirring speeds are shown in Table 3. The preparation procedure was as follows: firstly, oil-phase mixtures including T2t, T3t and T4t were prepared according to the description in Section 4.2.1 and formulas in Table 2. Secondly, the T3t mixture was transferred to a 55 °C preheated xanthan gum water solution; the xanthan gum concentration and water volume are shown in Table 3 (an expanded table including the T2t and T4t oil-phase mixtures is shown in Appendix A). Thirdly, the oil/water mixture was stirred at 55 °C for 3 min for emulsion; the stirring speeds were set at 2000, 6000 and 10,000 rpm, as shown in Table 3. The emulsion was transferred to an ice/water bath and stirred at 2000 rpm for 10 min to obtain the final AZO-loaded suspension. The AZO concentrations were 5% (wt/v) in all AZO-loaded suspensions. Two negative controls were prepared with the oil-phase mixture by removing AZO, AZO and DEP from T3t, according to the descriptions in Table 3, at the same time. The TDA-omitted suspension was named CK0. The particle sizes of the samples made in this step were measured.

### 4.3. Preparation of AZO-Loaded NPs with Ionic Emulsifiers and Optimization of O/W Ratios 

The anionic emulsifier SDS and cationic emulsifier CTAB were added to the AZO-loaded nanosuspension to endow negative or positive zeta potentials to the AZO-loaded NPs for improvement of stability via the intensification of electronic repulsion among particles. The optimization of the emulsifier and oil/water ratio was conducted based on the formula and protocol for T3t-13 AZO-loaded NPs. The emulsifier and oil/water ratios are shown in Table 4.

### 4.4. Characterization of the Particle Sizes, Zeta Potentials and Morphologies of AZO-Loaded NPs

The average particle size and zeta potential were determined using a Particle Zetasizer 3000HS at 25 °C. Prior to determination, the suspensions were diluted 1000 times. Measurements were performed in triplicate to obtain mean values and standard deviations. The FTIR analysis was performed using the spectrometer Paragon 1000 with KBr pellet in the range of 500–4000 cm^−1^. The DSC was performed using the calorimeter Pyris 1 in the range of −20 °C–150 °C at a heating rate of 20 °C min^−1^, under nitrogen gas protection. The TGA was performed using a thermal analysis system TGA 2 in the range of 28 °C–600 °C at a heating rate of 10 K min^−1^ under nitrogen gas protection. The XRD was performed on an x-ray powder diffractometer DX-2700 in the 2θ angle range of 5°–45° at 40 kV and 40 mA with an angular increment of 20° min^−1^. For the morphology observation, optimal NPs suspensions were diluted and dropped on 2 silicon slices for each sample, which were air dried at 20 °C and 30 °C, respectively. Then both slices for each sample were observed under a SEM SU1510.

### 4.5. Antifungal Activity Evaluation of Thermo-Responsive AZO NPs

The tested strains were the plant fungal pathogen *Phytophthora nicotiana* PNgz07 and the model fungus *Aspergillus niger* A1513. The *P. nicotiana* PNgz07 strain can infect the roots and stem shank of plants, causing black shank and death. The *A. niger* A1513 strain can infect cotton and juicy fruits to cause cotton bolls and fruit rot. The suppression efficacy of AZO-loaded NPs on the germination of A1513 spores and the growth of PNgz07 mycelium were determined as described below. Methods for the culture of PNgz07 and A1513 are described in the Appendix A.

#### 4.5.1. Suppressing Efficacy of AZO-Loaded NPs on the Germination of A1513 Spores

A1513 spores were collected from PDA agar and suspended in distilled water to obtain an A1513 spore suspension at a concentration of around 2 × 10^3^ cfu/mL. A gradient of T3t-13 and T3t-c12 AZO-loaded NPs was added to A1513 spore solutions at final AZO concentrations of 100 μg/plate, 80 μg/plate, 60 μg/plate, 40 μg/plate, 20 μg/plate and 0 μg/plate (as shown in Appendix A). A1513 spore solutions from AT3-100 and AT3-c100 were dropped on 2 silicon slices for each sample; then, the 2 slices were dried at 20 °C and 30 °C, respectively. The morphology of the spores was observed under SEM. The suppression rate of A1513 spore germination (Rsa, %) was calculated according to Equation (1). A correlation function between AZO concentration and lethal rate was derived and used for LC_50_ calculation:Rsa (%) = (1 − c1/c2) × 100% (1)

In Equation (1), Rsa (%) is the suppression rate of A1513 spore germination, c1 is the mean count of A1513 regenerating colonies per plate on plates containing AZO and c2 is the mean count of A1513 regenerating colonies per plate on negative control plates containing no AZO. 

#### 4.5.2. Suppression Efficacy of AZO-Loaded NPs on the Growth of PNgz07 Mycelium

T3t-c12 NPs were added to Tab-5 medium at a final AZO concentration of 100 μg/mL, while no AZO was added to the agar on the negative control plates (PN-20). After inoculation with PNgz07, PN-20 plates were cultured at 20 °C in a 12 h light/12 h dark cycle; the T3t-c12-added plates were separated into 2 groups and cultured with the same light/dark cycle at 20 °C and 30 °C, respectively. The diameters of PNgz07 colonies on agar were measured after 10 days of culture. Growth suppression rates (Rgs, %) were calculated according to Equation (2). The quantification of AZO residues in Tab-5 medium was performed according to a modified reverse-phase HPLC method [52,53]. Residual-free AZO in agar was extracted by mixing 2 g agar medium with 2 mL methanol thoroughly at room temperature; the supernatant was collected for HPLC determination. The HPLC running conditions were set as follows: acetonitrile: water: acetic acid = 50:50:0.4 (*v*/*v*/*v*) as the mobile phase; flow rate of 1 mL/min; UV detection wavelength at 245nm; and column temperature at 35 °C. The accumulative release percentage of AZO in agar was calculated by dividing the measured AZO concentration by the nominal concentration of AZO at the initial point (0d).
Rgs (%) = (1 − d1/d2) × 100%(2)

In Equation (2), Rgs (%) is the growth suppression rates, d1 is the mean diameter of colonies on the plates containing AZO, and d2 is the mean diameter of colonies on the negative control plates containing no AZO.

## 5. Conclusions

For the formation of thermo-responsive AZO-loaded NPs, the combination of TDA, DEP and T80—especially TDA and DEP—played a leading role by decreasing the softening point, interfering with crystallization during emulsification and assisting in the recovery of original AZO crystals after drying. The optimal thermo-responsive AZO-loaded T3t-c12 NPs contained 10% AZO (wt/v); had a mean particle size and zeta potential of 162.1nm and 5.1mV, respectively; whilst showed thermal response in a narrow warm temperature region between 20 °C and 30 °C; and possessed improved fungicidal efficacy against *A. niger* A1513 and *P. nicotiana* PNgz07. This unique thermo-responsive property in narrow warm temperature regions can be specified as a “Warm-Responsive Releasing (WRR)” property and has huge potential to cater to antimicrobial requirements during agricultural production. 

## Figures and Tables

**Figure 1 molecules-27-07959-f001:**
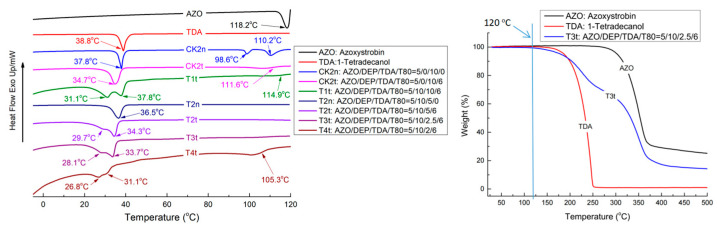
Thermal behaviors of oil-phase samples during DSC and TGA analysis.

**Figure 2 molecules-27-07959-f002:**
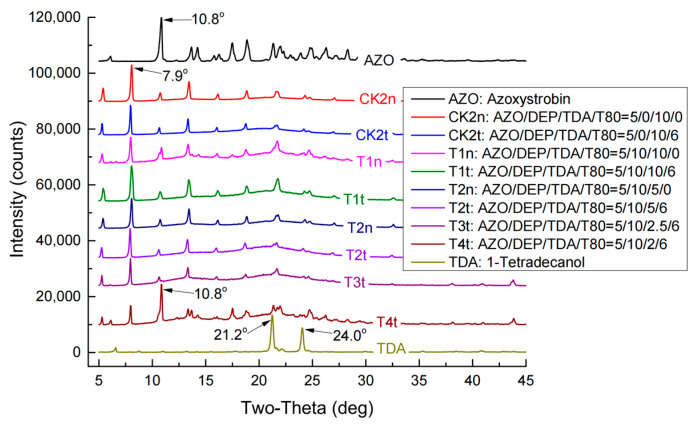
XRD patterns of oil-phase samples.

**Figure 3 molecules-27-07959-f003:**
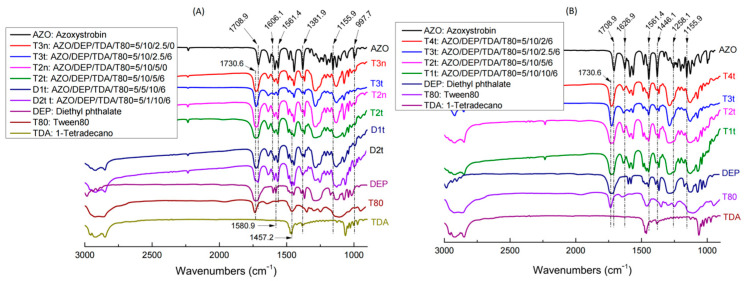
Comparison of FTIR peaks from oil-phase samples. The influences of DEP and T80 are shown in (**A**), and the effect of TDA is shown in (**B**).

**Figure 4 molecules-27-07959-f004:**
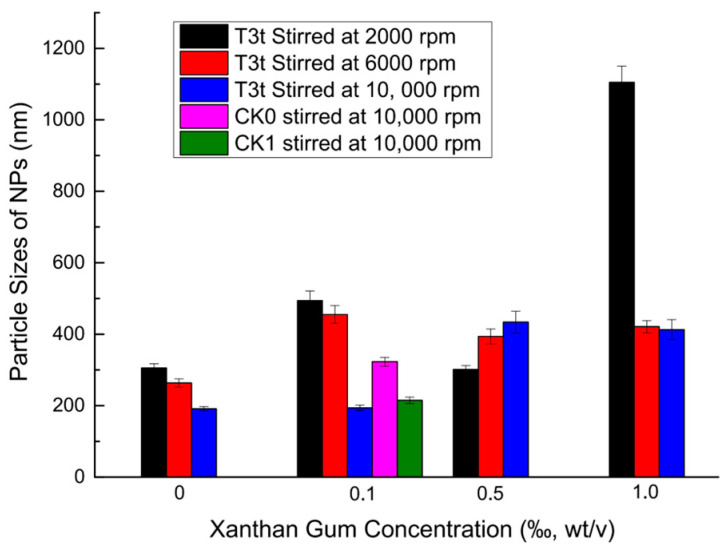
Particle sizes of AZO-loading NPs prepared with nonionic emulsifier (T80).

**Figure 5 molecules-27-07959-f005:**
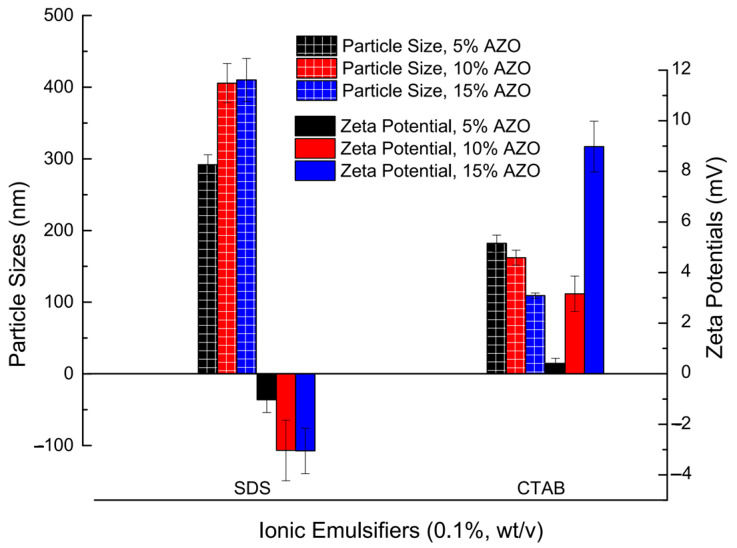
Particle sizes and zeta potentials of AZO-loaded NPs prepared with ionic emulsifiers.

**Figure 6 molecules-27-07959-f006:**
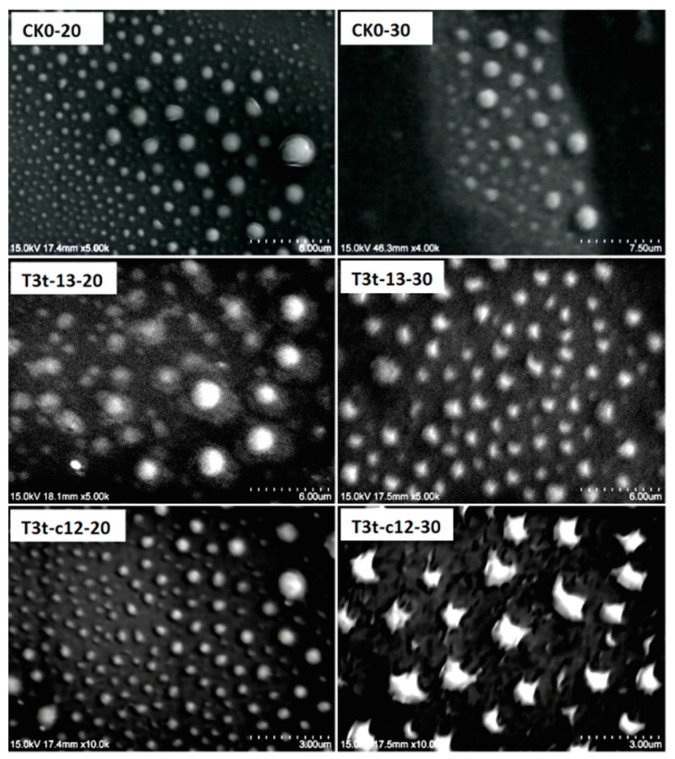
Morphology of dry particles from thermo-responsive AZO NPs under SEM. CK0-20 and CK0-30 were TDA-omitted AZO-loaded NPs desiccated at 20 °C and 30 °C, respectively; T3t-13-20 and T3t-c12-20 were NPs air-dried at 20 °C originating from T3t-13 and T3t-c12, respectively; T3t-13-30 and T3t-c12-30 were NPs air-dried at 30 °C originating from T3t-13 and T3t-c12, respectively.

**Figure 7 molecules-27-07959-f007:**
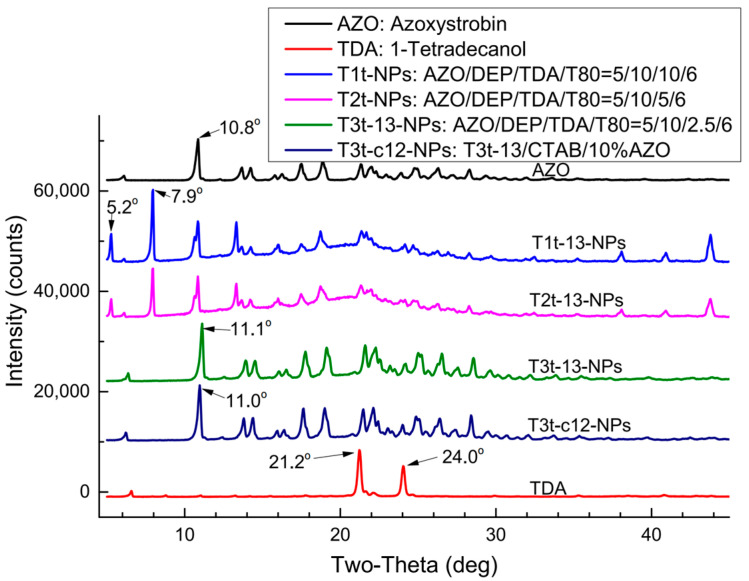
XRD patterns of AZO-TDA-DEP-T80 NPs. AZO and TDA represent the pure components AZO and TDA; T1t-13-NPs, T2t-13-NPs, T3t-13-NPs and T3t-c12-NPs were the dry NPs of T1t-13, T2t-13, T3t-13 and T3t-c12, respectively.

**Figure 8 molecules-27-07959-f008:**
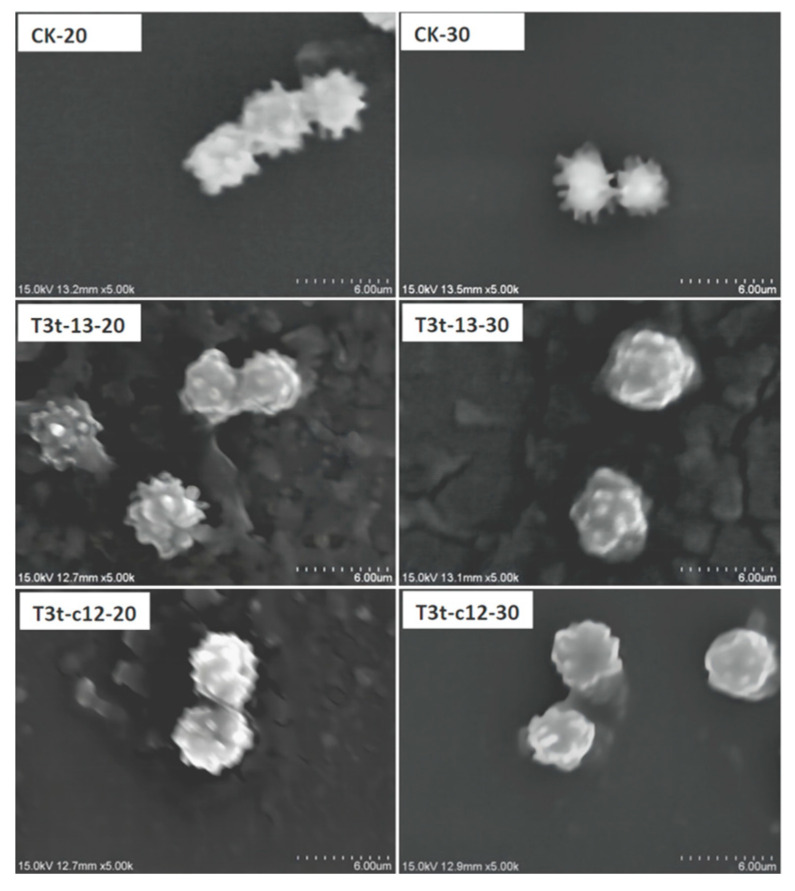
Morphology of *A. niger* A1513 spores with thermo-responsive AZO-loaded NP treatment under SEM. CK-20 and CK-30 were *A. niger* A1513 spores air-dried at 20 °C and 30 °C, respectively; T3t-13-20 and T3t-13-30 were *A. niger* spores air-dried at 20 °C and 30 °C, respectively, after co-incubation with T3t-13 NPs; T3t-c2-20 and T3t-c2-30 were *A. niger* spores air-dried at 20 °C and 30 °C, respectively, after co-incubation with T3t-c2 NPs.

**Figure 9 molecules-27-07959-f009:**
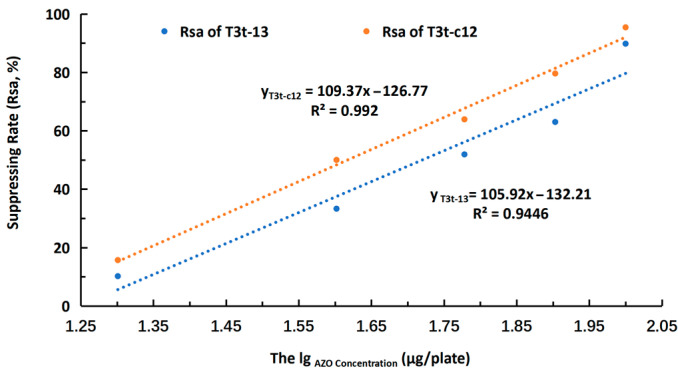
Correlation curve of AZO concentration and Rsa.

**Figure 10 molecules-27-07959-f010:**
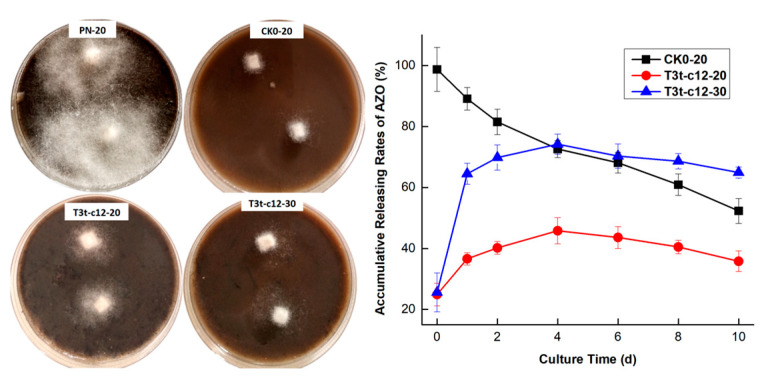
Mycelium of *Phytophthora nicotiana* PNgz07 on Tab-5 agar and accumulative release rate of AZO. Plate labeled PN-20 contained no AZO within the agar and was cultured at 20 °C (negative control); plates CK0-20 and T3t-c12-20 contained CK0 or T3t-c12 AZO-loaded NPs within the agar and were cultured at 20 °C; T3t-c12-30 contained the T3t-c12 AZO-loaded NPs within the agar and was cultured at 30 °C.

**Table 1 molecules-27-07959-t001:** Softening points and DSC endothermic peaks of oil-phase mixtures.

Sample Name	Softening Points (°C)	Endothermic Peaks in DSC (°C)
CK2t	36.1	34.7, 111.6
T1t	37.5	31.1, 37.8, 114.9
T2t	35.4	29.7, 34.3
T3t	31.5	28.1, 33.7
T4t	28.7	26.8, 31.1, 105.3
D1t	39.2	N/A
D2t	39.4	N/A
T5t	40.5	N/A

**Table 2 molecules-27-07959-t002:** Formula design for the screening of TDA and DEP additive amounts.

Sample Names	AZO/g	DEP/mL	TDA/g	T80/mL
CK2n	5	0	10	0
CK2t	5	0	10	6
T1n	5	10	10	0
T1t	5	10	10	6
T2n	5	10	5	0
T2t	5	10	5	6
T3n	5	10	2.5	0
T3t	5	10	2.5	6
T4n	5	10	2	0
T4t	5	10	2	6
D1n	5	5	10	0
D1t	5	5	10	6
D2n	5	1	10	0
D2t	5	1	10	6
T5t	5	5	5	6

**Table 3 molecules-27-07959-t003:** Xanthan gum concentrations and stirring speeds for AZO-loaded NP preparation.

Sample Names	Xanthan Gum (‰, wt/v)	Stirring Speeds (rpm × 1000)	Xanthan Gum Water Solution (mL)
T3t-01	0	2.0	81.5
T3t-02	0	6.0	81.5
T3t-03	0	10.0	81.5
T3t-11	0.1	2.0	81.5
T3t-12	0.1	6.0	81.5
T3t-13	0.1	10.0	81.5
T3t-21	0.5	2.0	81.5
T3t-22	0.5	6.0	81.5
T3t-23	0.5	10.0	81.5
T3t-31	1.0	2.0	81.5
T3t-32	1.0	6.0	81.5
T3t-33	1.0	10.0	81.5
CK0	0.1	10.0	82.0

**Table 4 molecules-27-07959-t004:** Emulsifier and oil/water ratio for AZO-loaded NPs optimization.

Sample Names	SDS (%, wt/v)	CTAB (%, wt/v)	Surfactant Water Solution (mL)	AZO Concentration (%, wt/v)
T3t-s11	0.1	0	81.5	5.0
T3t-s12	0.1	0	31.5	10.0
T3t-s13	0.1	0	14.8	15.0
T3t-c11	0	0.1	81.5	5.0
T3t-c12	0	0.1	31.5	10.0
T3t-c13	0	0.1	14.8	15.0

## Data Availability

Not applicable.

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
