# Peer review of "1-Tetradecanol, Diethyl Phthalate and Tween 80 Assist in the Formation of Thermo-Responsive Azoxystrobin Nanoparticles"

_molecules, 2022, doi:10.3390/molecules27227959_

Round 1
Reviewer 1 Report
In the manuscript entitled “The 1-Tetradecanol, Diethyl Phthalate and Tween 80 Procured Formation of Thermal-Response Azoxystrobin Nanoparticles by Downregulating Softerning Point of Oil Phase and Suppressing Crystallization of AZO”, G. Lin et al. have prepared thermo-responsive Azoxystrobin nanoparticles for agrochemicals applications.
In my opinion, the abstract is too long; please, reduce it reporting the objectives and the best obtained results of their reported study.
In the introduction, at line 83 page 2, the authors assert “no thermal-response AZO NPs has been reported”; it is not correct, please, modify.
The authors use poor comprehensible sample name, as reported in Table 1. Please, use a different and more simple and direct method to refer to samples, e.g. correlated to the sample preparation/characteristics, etc.
In the experimental section, the authors should give information about the type and model of used instrumentations, indicating also the conditions used to perform the analyses.
Improve the image quality of XRD patterns, FTIR spectra and SEM images; in Figure 3 which is the difference between spectra in A and in B. Explain and discuss.
Which is the investigated thermal range for the response? It is not clear, please, discuss. Does any chemical modification occur in the thermal variation? Please, discuss.
The English style should be improved.
I can accept with major revisions.
Author Response
Riviewer1:
In the manuscript entitled “The 1-Tetradecanol, Diethyl Phthalate and Tween 80 Procured Formation of Thermal-Response Azoxystrobin Nanoparticles by Downregulating Softerning Point of Oil Phase and Suppressing Crystallization of AZO”, G. Lin et al. have prepared thermo-responsive Azoxystrobin nanoparticles for agrochemicals applications.
(1)In my opinion, the abstract is too long; please, reduce it reporting the objectives and the best obtained results of their reported study.
Response: the abstract was reduced according to this suggestion. The length of abstract was shortened from 222 words to 200 words.
First sentence was rewritten and shortened.
Second sentence was modified to direction statement of objective. The previous sentence is: “The fabrication of thermal response agrochemical nanoparticles could help to integrate the small dimension effect and smart release performance in response to environmental stimuli, improve efficacy and reduce adverse effects.”
The rewritten sentence is:” To get antifungal agrochemicals released and fungicidal efficacy improved in response to warm condition, series of thermal-responsive agrochemical-loaded nanoparticles were created.”
(2)In the introduction, at line 83 page 2, the authors assert “no thermal-response AZO NPs has been reported”; it is not correct, please, modify.
Response: Thanks a lot for this kind remind, the “no thermal-response AZO NPs has been reported” was deleted.
We search again on “web of science” and other platform with key words “thermal and azoxystrobin”. some reports containing results about thermal stability of azoxystrobin and stability of azoxystrobin particles under different thermal conditions were confirmed and integrated in introduction.
(3)The authors use poor comprehensible sample name, as reported in Table 1. Please, use a different and more simple and direct method to refer to samples, e.g. correlated to the sample preparation/characteristics, etc.
Response: shall we maintain the present sample names, while add sample descriptions including information of components and preparation methods to figures?
We ever used the names which were correlated to sample components and preparation conditions. But because that there were 6 factors optimized for preparation of AZO-loaded thermal response nanoparticles. So, the previos names were really long. For example, T3t-c12 had a previous name as “AZO10@TDA2.5-T80-XG0.1-ST6k-CTAB0.1”. Then we had to switch to shorter names as you see in the manuscript now. In current naming rule of our samples, the “T” in T3t-c12 refers the “T” both from “Thermal” and “TDA”; the “3” is for the 3rd TDA amount of 2.5g; “c” refers CTAB; “12” refers level 1st xanthan gum concentration of 0.1 and 2nd stiring speed of 6000rpm. In addition,the current naming rule is consistent throughout the whole manuscript. The oil phase of T3t-c12 was named as T3t in Table1, nanosuspension without CTAB was named as T3t-12 in Table 2; after CTAB plus, it was named as T3t-c12 in Table 3.
In the end, this recommendation make us realize that the current short names may be not friendly to readers because of their poor comprehensible shortcomings. As a remedy, we added sample descriptions including information of components and preparation methods to figures. Thus, readers will be able to catch the detailed information of samples at their first glance on figures.
(4)In the experimental section, the authors should give information about the type and model of used instrumentations, indicating also the conditions used to perform the analyses.
Response: Informations about the type and model and working conditions of instrumentations have been added to the experimental section.
(5)Improve the image quality of XRD patterns, FTIR spectra and SEM images; in Figure 3 which is the difference between spectra in A and in B. Explain and discuss.
Response to the question about chemical modification
Resolutions of images have been improved.
Response to the question about Figure 3-A and -B
Figure 3-A showed the effect of Tween80 and DEP on FTIR results. We added “Figure 3-A” in this graph to point out results from Figure 3-A.
***due to the AZO-induced co-frequency phenyl vibration of DEP (Figure 3-A). According to decreasing DEP amounts (from T2t to D2t to D1t in Figure 3-A), four variations in vibration peaks were also found:
Figure 3-B showed the effect of TDA amounts on FTIR results. There is a sentence to explain this: “The effect of TDA amounts on FTIR results was shown in Figure 3-B” in this graph.
(6)Which is the investigated thermal range for the response? It is not clear, please, discuss. Does any chemical modification occur in the thermal variation? Please, discuss.
Response to the question about thermal range:
The thermal range for the response is 20 oC to 30 oC. Our previous expression should be a little bit fuzzy and vague, so we changed the statement about thermal response according to this question as follow.the proposed oil phase with ability of form switch from being solid at 20 oC to be softerning at 31.5 oC. thermal response morphological transformation between 20 oC and 30 oC,***showed satisfied antifungal efficacy against P. nicotiana PN07 and A. niger A1513 at 30 oC. Thus, thermal range was introduced clearly.
Response to the question about chemical modification
There should no detective chemical modification in the thermal variation range of 20 oC - 30 oC because the temperature located at a mild range and all components in the AZO-loaded system have nice stability.
(7)The English style should be improved.
The manuscript has been subjected for English edition to MDPI's English editing service.
Thanks a lot for all questions and recommendations.
Yong Liu and coauthors
Reviewer 2 Report
In this manuscript, authors reported the preparation of thermal-response AZO-loaded NPs for antifungal purposes. They conducted a series of experiment that consist of a screen of an AZO oil-phase formula with softening point close to 30°C, accompanied with an optimization of emulsification conditions to prepare thermal-response AZO-loaded NPs. The manuscript presents some good and interesting data. I reviewed the manuscript in a critical manner and some of the comments are given below:
General comments
The manuscript might be a contribution of interest for “Molecules” and in principle within its specific scope, it is suitable for publication after addressing some minor issues. The manuscript is sufficiently organized with well stated objectives. The quality of writing is ‘more or less’ acceptable with some grammar and spelling especially in the introduction section. The English language usage should be checked by a fluent English speaker and/or a professional language editing service.
In addition, the introduction section needs to be more coherent, summarizing and describing the content of the full text, so it needs to enrich the content.
I recommend Acceptance after minor revision noted
Specific comments
1. Many studies in the state of the art used different kinds AZO-loaded NPs for antifungal purposes. Therefore, what is the novelty and the benefit of this work compared with those already reported? I suggest that authors highlight more the novelty in this study.
2. The introduction section is somewhat poor in terms of bibliographic data. A discussion is very well desired in the introduction section to highlight the state of the art recent advances in this kind of materials.
3. The title is very a long and confusing. I suggest a full reformulation of the title.
4. DSC interpretation needs to be improved. Authors tend to give observations (increase/decrease) and not interpretation (WHY the increase or decrease)
4. Authors are welcomed to provide TGA/DTG curves to highlight the thermal stability/decomposition of the prepared samples.
Overall, a very structured work with few points to critique.
Author Response
Reviewer 2
In this manuscript, authors reported the preparation of thermal-response AZO-loaded NPs for antifungal purposes. They conducted a series of experiment that consist of a screen of an AZO oil-phase formula with softening point close to 30°C, accompanied with an optimization of emulsification conditions to prepare thermal-response AZO-loaded NPs. The manuscript presents some good and interesting data. I reviewed the manuscript in a critical manner and some of the comments are given below:
General comments
The manuscript might be a contribution of interest for “Molecules” and in principle within its specific scope, it is suitable for publication after addressing some minor issues. The manuscript is sufficiently organized with well stated objectives. The quality of writing is ‘more or less’ acceptable with some grammar and spelling especially in the introduction section. The English language usage should be checked by a fluent English speaker and/or a professional language editing service.
In addition, the introduction section needs to be more coherent, summarizing and describing the content of the full text, so it needs to enrich the content.
I recommend Acceptance after minor revision noted
Specific comments
- Many studies in the state of the art used different kinds AZO-loaded NPs for antifungal purposes. Therefore, what is the novelty and the benefit of this work compared with those already reported? I suggest that authors highlight more the novelty in this study.
Response: The responsive releasing of AZO in a warm temperature range of 20 oC - 30 oC is main highlight point.
Our previous expression should be a little bit fuzzy and vague, so we changed the statement about thermal response in abstract according to this question as follow:
- the proposed oil phase with ability of form switch from being solid at 20 oC to be softerning at 31.5 o
(2) thermal response morphological transformation between 20 oC and 30 oC,
(3) showed higher antifungal efficacy against P. nicotiana PN07 and A. niger A1513 at 30 oC. Thus, the achievement of this paper was highlighted.
- The introduction section is somewhat poor in terms of bibliographic data. A discussion is very well desired in the introduction section to highlight the state of the art recent advances in this kind of materials.
Response: The introduction was supplemented with some new references and the discussion about highlight in materials also was reorganized.
The preparation of azoxystrobin nanopartilce were separated in two categories: AZO crystals and AZO-loaded composited nanoparticles. And preparations of AZO-loaded nanoparticles also were summarized as several kinds. The followings are 2 sentences for initiations of introduction of the two categories:
(1) To minimize the size of AZO crystals to nanoscale is one of the popular research purposes.
(2) In addition to preparation of nanoscale AZO crystals, a series of AZO-loaded composited nanoparticles also were reported.
Detailed modification can be seen in main manuscript.
- The title is very a long and confusing. I suggest a full reformulation of the title.
Response: The title was polished and shortened.
The previous title: The 1-Tetradecanol, Diethyl Phthalate and Tween 80 Procured Formation of Thermal-Response Azoxystrobin Nanoparticles by Downregulating Softerning Point of Oil Phase and Suppressing Crystallization of AZO
The simplified title: 1-Tetradecanol, Diethyl Phthalate and Tween 80 Assist in the Formation of Thermal Response Azoxystrobin Nanoparticles
4.DSC interpretation needs to be improved. Authors tend to give observations (increase/decrease) and not interpretation (WHY the increase or decrease)
Response: Several short discussions about DSC interpretation were added to DSC results part.
The phase transformations of TDA from metastable hexagonal orthorhombic solid phase (SHEX) to orthorhombic solid phase (SORT), and formation of defective TDA crystals are cited to explain the movements of endothermic peaks in Figure 1.
- 5.Authors are welcomed to provide TGA/DTG curves tohighlight the thermal stability/decomposition of the prepared samples.
Response: TGA was performed and the curve was inserted next to the DSC figure.
Main components and optimal formula are thermal stable in the temperature range (-10 oC - 120 oC) for DSC analysis.
Thanks a lot for all questions and recommendations.
Yong Liu and coauthors
Round 2
Reviewer 1 Report
In my opinion, the revised version of the manuscript can be accepted for publication.
Author Response
Dear reviewer,
Thanks a lot for your work on our manuscript.
Sincerely,
Yong Liu and coauthors